# Dispersion of Multi-Walled Carbon Nanotubes into White Cement Mortars: The Effect of Concentration and Surfactants

**DOI:** 10.3390/nano12061031

**Published:** 2022-03-21

**Authors:** Zoi S. Metaxa, Spyridoula Boutsioukou, Maria Amenta, Evangelos P. Favvas, Stavros K. Kourkoulis, Nikolaos D. Alexopoulos

**Affiliations:** 1Hephaestus Laboratory, Department of Chemistry, International Hellenic University, St. Luke, 65404 Kavala, Greece; zmetaxa@chem.ihu.gr (Z.S.M.); amentamari@gmail.com (M.A.); 2Research Unit of Advanced Materials, Department of Financial Engineering, School of Engineering, University of the Aegean, 41 Kountouriotou Str., 82132 Chios, Greece; irida.bou@gmail.com; 3Institute of Nanoscience and Nanotechnology, NCSR ‘‘Demokritos”, 15341 Agia Paraskevi, Attica, Greece; e.favvas@inn.demokritos.gr; 4Laboratory of Testing and Materials, Department of Mechanics, National Technical University of Athens, 9 Heroes Polytechniou Str., 15780 Athens, Greece; stakkour@central.ntua.gr

**Keywords:** cement-based materials, superplasticizer, MWCNTs, Aalborg cement, flexural strength, electrical resistance

## Abstract

Multi-wall carbon nanotubes (MWCNTs) exhibit exceptional mechanical and electrical properties and can be used to improve the mechanical and piezoelectric properties of cement-based materials. In the present study, the effect of different MWCNT concentrations as well as different types of surfactants and a superplasticizer were examined to reinforce, at the nanoscale, a white cement mortar typically used for the restoration of monuments of cultural heritage. It was shown that sodium dodecylbenzenesulfonate (SDBS) and Triton X-100 surfactants slightly decreased the white cement mortars’ electrical resistivity (by an average of 10%), however, the mechanical properties were essentially decreased by an average of 60%. The most suitable dispersion agent for the MWCNTs proved to be the superplasticizer Ceresit CC198, and its optimal concentration was investigated for different MWCNT concentrations. Carboxylation of the MWCNT surface with nitric acid did not improve the mechanical performance of the white cement nanocomposites. The parametric experimental study showed that the optimum combination of 0.8 wt% of cement superplasticizer and 0.2 wt% of cement MWCNTs resulted in a 60% decrease in the electrical resistivity; additionally, the flexural and compressive strengths were both increased by approximately 25% and 10%, respectively.

## 1. Introduction

Carbon nanotubes (CNTs) exhibit exceptional mechanical and electrical properties and can be used to improve the mechanical [1] and piezoelectric properties of cement-based materials [2]. There are multiple factors that determine the effect of the addition of CNTs on the properties of nano-reinforced cement-based materials. Two of the most important factors are the dispersion and the concentration of CNTs in the cementitious matrix [3,4].

CNTs tend to form agglomerates due to van der Waals forces. The methodology followed for the successful dispersion of CNTs is considered critical, as it allows for the nano-reinforced material to exhibit homogeneous electrical and mechanical properties. The dispersion methods proposed in the literature can be divided into two main categories: mechanical (or physical) methods and chemical ones [5], which are either used individually or in combination. Mechanical dispersion methods such as high-energy sonication are used to successfully separate CNT agglomerates. Chemical methods use superplasticizers or surfactants to improve their dispersion or adhesion to the matrix by surface modification, thus minimizing their agglomeration.

The most widely used method for the successful dispersion of CNTs is the ultrasonication technique [6]. This method is mainly used for the dispersion of CNTs in low viscosity fluids such as water, acetone, and ethanol [7,8]. The ultrasound device converts electrical voltage into mechanical vibrations. These mechanical vibrations are transmitted through a probe to the CNT suspension, creating pressure waves. These waves cause tiny bubbles to form and collapse. This phenomenon, called cavitation, leads to the formation of waves by increasing the temperature of the liquid [9]. The amount of energy released by each individual bubble is small, however, the cumulative effect results in extremely high energy levels in the suspension, leading to successful CNT dispersion. Other typical methods of dispersing nanoparticles in a matrix are the calendaring process, ball milling, the shear mixing method, the extrusion technique [8,10], etc. The roller technique leads to a fairly good dispersion of CNTs in an epoxy resin matrix [11] and has been widely used to produce nano-reinforced composites. The ball grinding technique has been used successfully; however, it has been shown in the literature [5,7] that it alters the structure of nano-reinforcements (e.g., reduces the length of nanotubes). The shear mixing method has been widely used to disperse CNTs into resins. Nevertheless, it has been observed that in several operations, the CNTs did not disperse satisfactorily, showing several agglomerates. Finally, the extrusion technique has been used to disperse CNTs into thermoplastic polymers [7,12], with poor results.

Chemical methods can be divided into two different categories. The first category includes those methods that modify the surface of CNTs chemically using covalent modification. The second category includes non-covalent treatment methods that modify the surface of CNTs [13,14]. In the first case, the CNT dispersion is in suspension in the presence of polymers such as polyphenylene vinylene (PPV) or polystyrene (PS) [15]. In this case, the polymer molecules are wrapped on the surface and around the CNTs. Various surfactants other than the polymers presented above have been used as dispersants. The different types of surfactants found in the literature include: (a) nonionic surfactants such as Triton-X100 [8,16,17]; (b) anionic surfactants such as sodium dodecyl sulfate (SDS), sodium dodecylbenzenesulfonate (SDBS), and polystyrene sulfate (PSS) [17,18]; and finally, (c) cationic surfactants such as dodecyl trimethyl ammonium bromide (DTAB) [18] and cetyltrimethylammounium 4-vinylbenzoate (CTVB). The efficiency of this method depends on both the properties of the surfactant itself and on its CNT suspension content [17]. Superplasticizers are a specific type of anionic surfactants regularly used to reduce the mixing water in concrete, improve the rheological properties, and improve watertightness. These materials have been successfully used for the dispersion of CNTs in cement-based materials [19].

The covalent modification (the carbon sp^2^ bond of the nanotubes is converted to a sp^3^ bond) is an alternative method for dispersing CNTs, which improves the chemical compatibility of the CNTs with the matrix. This increases the likelihood of good adhesion to the matrix and generally reduces their inherent ability to develop van der Waals forces, which cause agglomerates. This modification occurs when reacting with molecules or compounds of high chemical reactivity such as fluorine, amines, and various hydroxyl groups [7]. Several methods of modifying the surface of CNTs have been proposed in the literature, with the most common being the use of various acids (e.g., nitric acid or the combination of acids) to oxidize the surface and create functional groups such as carboxylic [5].

The incorporation of CNTs in cementitious matrices made from ordinary Portland cement has been widely studied [1]. In contrast, there is minimal research on their use in white cement matrices [20]. Research on white cement and the addition of nanomaterials has focused on microscopic techniques due to the color contrast between the white matrix and the black CNT agglomerates.

In the present work, the main factors that affect the performance of nano-reinforced cement mortars were studied. First, the dispersion methodology was examined using: (i) two surfactants frequently used for the dispersion of CNTs in polymers; (ii) a superplasticizer compatible with cement-based materials; and (iii) carboxyl-functionalization of the CNTs. Second, the effect of the CNT concentration in the cement matrix on the mechanical and electrical properties of the mortars was examined.

## 2. Materials and Methods

### 2.1. Materials

In the present study, a white Portland cement AALBORG WHITE CEM I 52,5R (Aalborg White, Aalborg, Denmark) was used. The mineralogical composition and main physical characteristics are presented in Table 1 and Table 2. A coarse grained quartz sand (1–2 mm) and a fine-grain quartz sand (Μ32, average grain size 260 μm) were used as aggregates. The selection of the raw materials was based on the various criteria discussed in [21,22].

MWCNTs were produced by Glonatech S.A., Lamia, Greece in a fluidized bed chemical vapor deposition vertical reactor having 94% purity, length >5 μm, and diameter ranging from 20 to 45 nm. For the successful dispersion of MWCNTs in the cement mortar matrix, the use of three commercially available surfactants was examined: (a) SDBS (sodium dodecyl benzene sulfonate); (b) Triton X-100; and (c) Ceresit CC198 (FM)/(BV).

Sodium dodecyl benzene sulfonate (SDBS) is an ionic surfactant that consists of a number of organic compounds and has the following chemical formula: C_12_H_25_C_6_H_4_SO_3_Na. It is a colorless salt widely used to disperse MWCNTs in aqueous solutions [17], polymeric matrices [23], and cementitious materials [24,25]. According to Duan et al. [26], carbon nanotubes are dispersed by the adsorption of surfactant molecules on MWCNTs. One of the objectives of the present research was to investigate the compatibility of the above surfactant with the cement-based materials as the use of ionic surfactants results in a decrease in the mechanical properties of the cement matrix due to the formation of foam during mixing of the cement paste [27].

The Triton X-100 (TX-100) solvent is a non-ionic surfactant, which, similar to SDBS, has been widely used to disperse MWCNTs in aqueous solutions [28,29]. Since it has not yet been determined which of the two surfactants, SDBS or Triton X-100, is more suitable for the dispersion of MWCNTs, the examination of both as possible MWCNT dispersing agents for cement-based materials was decided.

Ceresit CC198 (FM)/(BV) is a new generation polycarboxylate-ether superplasticizer that also contains a small amount of lignin sulfonate. It is fully compatible with cement and is widely used in construction, mainly to improve the workability of cement-based materials. The use of this type of superplasticizer has been reported in the literature to form a uniform dispersion of MWCNTs [30].

### 2.2. Preparation of Aqueous MWCNT Dispersions

MWCNT dispersions were prepared by mixing MWCNTs with the mixing water containing a surfactant. In order to achieve a homogeneous suspension, the mixtures were subjected to high ultrasonic energy at room temperature using a 200 W high-intensity ultrasonic device (UP200S) equipped with a 3 mm diameter cylindrical tip suitable for 200 mL dispersions. The ultrasound device operated at 50% of its power output for one hour. The device operated in cycles of 0.5 s to prevent overheating of the aqueous suspensions. The above suspensions were divided into three equal parts and the dispersion was carried out in turns for one hour. Finally, the suspensions were placed together in a suitable container and mixed by hand.

Table 3 shows the mixing quantities used for the development of the different MWCNT/aqueous suspensions. The SDBS surfactant was used for the dispersion of different MWCNT concentrations, at the same amount as MWCNTs (mixes 1–3) and double the amount of MWCNTs (mixture 4). Triton X-100 was also used to disperse the MWCNTs in proportions of 2.4% by weight of Triton X-100 in water and 0.4% by weight of MWCNTs in water (Triton X-100).

Subsequently, five dispersions of MWCNTs and Ceresit CC198 (FM)/(BV) superplasticizer were prepared. In all mixtures, the MWCNT concentration remained constant at 0.4 wt% of water (Table 3). Additionally, five suspensions were prepared to determine the effect of the MWCNT concentration on the dispersion quality. In all suspensions, Ceresit CC198 was used as the optimal type of solvent at its optimal concentration (1.6 wt% of water). The mixing ratios of the suspensions are shown in Table 3. Finally, based on the results of the five different MWCNT contents, the content (0.4 wt% of water MWCNTs) was selected for which MWCNT chemical modification was performed. Chemical covalent functionalization of MWCNTs was performed by attaching carboxylic groups to the CNTs. CNTs were agitated with 65% nitric acid over a temperature range of 70–150 °C for 24 h under mechanical agitation. The solution was then rinsed to remove the excess acid until the pH became neutral. Finally, it was dried in a vacuum oven. After their modification, the MWCNTs (suspension MWCNTs-6) were dispersed in water containing the Ceresit CC198 surfactant using high energy ultrasonication.

The two types of MWCNTs investigated were explicitly analyzed in a preliminary article of the authors [31]. The preliminary performed X-Ray Photoelectron Spectroscopy (XPS) measurements showed that after the treatment process, the C presentence decreases and the sp^2^/sp^3^ ration increases, which indicates that the material after the treatment contains carbon atoms, mostly in graphite-like sp^2^ and less in diamond-like sp^3^ hybridization states. This increase in the sp^2^, C=C layers spacing, is also supported by the main intense X-ray diffraction (XRD) peak at 2θ = 26.3°. Compared to the normal graphite, 2θ = 26.5°, this peak showed a downward shift.

### 2.3. Preparation of Cement Mortar Specimens

The flow diagram of the current investigation is summarized in Figure 1; initially the MWCNTs were dispersed in the aqueous solution following the aforementioned method. The resulting MWCNT suspension was mixed with cement and sand in a typical cement mixer. After mixing, the mortar was poured into molds for casting prismatic and cylindrical specimens for electrical resistivity measurements, flexural as well as compression tests, respectively.

Mixing was performed according to the ASTM C305 standard using a standard 5 L mixer from TECHNOTEST^®^ (Treviolo, Itally). First, water or MWCNT suspensions were added to the cement and mixed for 30 s at slow speed (140 ± 5 r/min). Second, premixed aggregates (coarse and fine) were added into the mixing bowl over a 30 s period while mixing at slow speed. Then, the mixing was continued for another 30 s at higher speed (285 ± 10 r/min). Next, a resting period of 90 s was implemented in order to let the mixture stand. At this time, the mixture was scraped from the sides of the bowl and mixed into the batch. Finally, the mixing was resumed for another 90 s at medium speed to ensure the mixture’s homogeneity. The mix proportions for the powder materials are shown in Table 4. The water to cement ratio (*w*/*c*) was kept constant for all samples at 0.5.

Customized prismatic molds were manufactured with a length of 80 mm and a cross-section of 20 mm × 20 mm. These molds were used for casting specimens for electrical resistivity measurements as well as flexural testing. For the electrical resistivity specimens, four steel electrodes that covered the entire cross section of the specimens were inserted inside the samples immediately after casting to support the four-wire method for electrical resistance measurement. The distance between the outer and inner electrodes was 15 mm and between the inner electrodes was 30 mm.

Finally, cylindrical molds with dimensions 30 mm × 60 mm were used to cast the specimens for compression tests. Before casting, suitable concrete demolding oil was used to facilitate the removal of the specimens. The specimens were subsequently cured in calcium hydroxide saturated water for 28 days.

The different experimental batches produced in the present study can be seen in Figure 2. Initially, the investigation included the optimal surfactant type (experimental batches 1–3). Following this, using the optimum surfactant type and its concentration, respectively, the effect of the MWCNT concentration was investigated (experimental batch 4). Finally, the possible effect of MWCNT carboxylation was studied (experimental batch 5).

### 2.4. Electrical Resistance Measurements and Mechanical Testing

After 28 days of curing, the specimens were removed from the hydration tanks and rinsed under running water. Following this, they were placed in a drying oven for three days at 80 °C to remove the free water to avoid any possible polarization effects during the measurements. The drying oven had a temperature control with an accuracy of ±0.1 °C. The specimens, after their removal from the oven, were allowed to return to room temperature for about 30 min and then their dimensions were measured.

An Agilent 34970a multimeter (Santa Clara, CA, USA) was used in situ to record the electrical resistance data of the specimens through the embedded metallic grids, Figure 3a. The resistance measurements were performed in a four-point measurement set up in the longitudinal direction following Ohm’s law. The internal electrodes were used to measure the voltage, while the outer electrodes provided a continuous supply of constant current. The electrical resistivity was calculated according to Ohm’s law taking into consideration the specimens’ electrical resistance and external dimensions (see Figure 3a). Data acquisition of 1 Hz was used for the resistance measurements, while the data were simultaneously transmitted in the P/C of the testing machine for a minimum of 30 min. The electrical resistivity of the samples was determined as the mean value of the electrical resistivity values during the last five minutes (25 up to 30 min) of testing.

Flexural strength tests were performed in an MTS Insight (Eden Prairie, MN, USA) 10 kN loading frame (Figure 3b). Before testing, an artificial notch, 6 mm in depth and 2 mm in width, was introduced at half width of the specimens. Crack width was measured using a crack mouth opening displacement (CMOD) extensometer (Instron, Norwood, MA, USA). The tests were performed following the 3-point bending test set up with a spam length of 70 mm and a displacement rate of 0.001 mm/s.

A 300 kN Instron SATEC loading frame (Norwood, MA, USA) was used for the compressive strength tests with a displacement rate of 0.3 mm/min. Load, displacement, time, and crack mouth opening displacement (CMOD) measurements were recorded during testing. Three specimens were tested from each mixture and the average strength value was calculated. In total, more than 60 specimens were used for electrical resistivity, while more than 50 specimens were tested for flexural tests and more than 50 specimens were tested in compression.

## 3. Results and Discussion

The experimental test results are presented and discussed in this section in order to establish a useful correlation of the MWCNT dispersion, the electrical resistivity, and the mechanical performance of the produced specimens.

### 3.1. Effect of the Surfactant Type

To efficiently disperse the MWCNTs, several surfactants were exploited to prepare/modify the surface structure of the nano-reinforcements with non-covalent functionalization to increase the dispersion state and the interface bonding with the surrounding matrix. Two types of surfactants were used in the present work, one ionic (SDBS) and one non-ionic (Triton X-100), which have been widely used in the dispersion of such reinforcements in resin matrices [32,33].

Figure 4a depicts the typical electrical resistivity curves of the MWCNT-reinforced white cement mortar specimens; the specific MWCNT concentrations were dispersed using the SDBS and Triton X-100 surfactants, respectively. Electrical resistivity of the reference (no MWCNTs reinforced) cement mortar exhibited resistivity values around 1 ΜOhm·cm with increased amplitude/variability of the experimental measurements. The addition of both surfactants decreased the amplitude of the measurements, nevertheless, the improvement in electrical properties remained unclear for the case of the SDBS surfactant. In Figure 4b, the average values with the respective standard deviation of the experimental electrical resistivity measurements are shown. A small improvement (reduction) could possibly be noticed for the low SDBS concentrations. Addition of the Triton X-100 surfactant seems to be beneficial for the electrical resistivity values since almost 20% lower electrical resistivity was measured. As a general observation, the use of these two surfactants slightly decreased the electrical resistivity, especially for the low concentration specimens. Nevertheless, this decrease was marginal and within the standard deviation of the measurements.

Typical flexural stress–crack mouth opening displacement curves for the investigated surfactants to disperse the MWCNTs can be seen in Figure 5a. It seems that all investigated cases decreased the flexural strength (peak flexural stress) of the specimens and generally a more brittle behavior was noticed. Particularly for the 0.25 and 1.0 wt% SDBS specimens, all samples were fractured immediately after the initiation of the mechanical testing and therefore their flexural mechanical behavior was not recorded and presented. Similarly, in the case of the Triton X-100 surfactant, large pores were noticed, and the specimens were almost destroyed after demolding (Figure 6). To this end, the lowest flexural strength of less than 1 MPa does not seem to be a surprising result.

Figure 5b shows the flexural strength results of the investigated surfactant concentrations in terms of the average values and respective standard deviation. All investigated concentrations essentially decreased the flexural strength of the specimens, when compared against the reference cement mortar. The use of the SDBS surfactant in high concentrations resulted in embrittlement of the specimens. In particular, for the Triton X-100 surfactant, flexural strength was decreased by nearly 80%. The compression strength results were also plotted in the same figure, showing that these surfactants also affected the mechanical behavior of the specimens under compressive loads. In all investigated cases, compressive strength was essentially decreased. The observed reduction in the mechanical properties accompanied with the use of non-ionic surfactants such as SDBS and Triton X-100 is related to the formation of foam during the mixing process. As a result, air is entrapped in the cementitious matrix. According to the literature, this leads to a material with lower bulk density, increased porosity, and poor mechanical performance [34,35,36,37,38,39]. The results indicate that the exploitation of these two surfactants (SDBS and Triton X-100) to disperse MWCNTs in white cement mortars should be avoided as it has been proven that they are not compatible with the cementitious matrix.

### 3.2. Effect of the Superplasticizer Concentration

The use of a commercially available superplasticizer was investigated for the efficient dispersion of the MWCNTs before mixing in the white cement mortar matrix, since the previously investigated surfactants did not succeed in achieving increased mechanical properties in the white cement mortar. This was due to morphological differences (i.e., increased porosity) that are evident even at the macroscale (Figure 7) when looking at the samples’ outer surface. The MWCNT/white cement mortar dispersed using a superplasticizer demonstrated a smoother outer surface. In contrast, the samples with the SDBS surfactant had large macroscopic pores.

Superplasticizers are surfactants typically used in cementitious materials to improve their workability and normally do not influence the mechanical properties of the cementitious matrix [40,41]. To this end, a typical concentration of 0.2 wt% MWCNTs was selected from the literature [42,43] as a constant concentration to investigate the effect of superplasticizer concentration on the MWCNT dispersion. Figure 8a shows the typical electrical resistivity curves of the investigated mortars reinforced with a 0.2 wt% MWCNT concentration; the MWCNTs were mixed in solutions that had various superplasticizer concentrations. The addition of MWCNTs decreased the electrical resistivity values; the 0.8 wt% superplasticizer concentration resulted in the lowest measured electrical resistivity value. Average values and standard deviation of the average electrical resistivity values can be seen in Figure 8b for the various superplasticizer concentrations. The available experimental test results were connected with the aid of a B-Spline curve (eye-catch) for the convenience of the reader. It is evident that electrical resistivity values decreased with an increase in the superplasticizer concentration up to the minimum observed for the case of 0.8 wt% Ceresit concentration, indicating that lower surfactant concentrations are not sufficient to homogeneously disperse the MWCNTs. For this specific concentration (0.8 wt% of cement), the standard deviation of the experimental measurements was the lowest measured, thus proving the high reproducibility among the different investigated specimens of the same batch as well as the high dispersion condition of the MWCNTs. Higher superplasticizer concentration (>0.8 wt%) led to a significant increase in electrical resistivity. This suggests that the MWCNT dispersion in the matrix was not efficient, as an excess amount of the Ceresit CC 198 superplasticizer caused flocculation of the surfactant molecules.

Figure 9a shows typical flexural stress–crack mouth opening displacement curves for the investigated superplasticizer concentrations. It is obvious that the superplasticizer concentration influences the flexural behavior of the MWCNT reinforced white cement mortar specimens. Small concentrations seemed to slightly increase the flexural behavior of the specimens; the 0.8 wt% concentration exhibited the highest increase in the peak value of the flexural curve. Higher concentrations (>1.2 wt%) decreased the maximum values of the respective curves, which might be caused by the insufficient nano-reinforcement dispersion.

Average values of flexural strength as well as the respective standard deviation for the various superplasticizer concentrations can be seen in Figure 9b. The results indicate that the superplasticizer affects the flexural behavior of cement mortar and its concentration is critical for the efficient dispersion of the MWCNTs in the cement matrix. The 0.8 wt% concentration specimens presented high flexural strength with extremely low standard deviation; the latter is evidence of the reproducibility of the experimental test results produced from the same test batch. Higher superplasticizer concentration did not seem to have a profound role on the MWCNT dispersion, and therefore presented slightly higher flexural strength values than the reference white cement mortar specimens. Nevertheless, the extremely low scatter in the flexural strength values suggests the stability and reproducibility of the produced specimens as well as of the experimental measurements.

Compression strength values can also be seen in Figure 8b in green filled squares and a dotted line (B-Spline curve, eye-catch) was also added to connect the available results. It was confirmed that the 0.8 wt% concentration provided the highest compressive strength as well as the lowest scatter within the experimental protocol investigating the effect of the superplasticizer concentration. According to the literature, the optimum superplasticizer to nanoscale fiber ratio to uniformly disperse both MWCNTs and carbon nanofibers (CNFs) for ordinary Portland cement is close to 4.0 [44,45]. This was also confirmed by this study for white cement mortars.

### 3.3. Effect of MWCNT Concentration

Figure 10a shows the typical electrical resistivity experimental curves of the mortar specimens for the various MWCNT concentrations investigated. It is evident that almost all the additions decreased the electrical resistivity of the white cement mortar matrix. The lowest value of electrical resistivity was measured for a 0.2 wt% MWCNT concentration, possibly due to the development of a more uniform MWCNT distribution network. The electrical resistivity results with average values and standard deviation for the investigated MWCNT concentrations can be seen in Figure 10b. All specimens presented low scatter in electrical resistivity and an eye-catch dotted line was also plotted. Electrical resistivity seemed to decrease up to a 0.2 wt% concentration, while *ρ* increased for higher concentrations.

Typical experimental curves of flexural stress–CMOD can be seen in Figure 11a for the investigated MWCNT concentrations. It is evident that all MWCNT additions had a profound effect on the flexural strength of the specimens, with the optimal concentration being around 0.2 wt%. Lower concentrations could not sufficiently reinforce the matrix, while non-uniform dispersion of the reinforcement might be the reason for the flexural strength decrease for higher concentrations.

Figure 11b shows the flexural strength results of the investigated MWCNT concentrations in terms of average values and respective standard deviation. Concentrations of 0.1 and 0.3 wt% also exhibited high flexural strength values, nevertheless, high scatter (standard deviation) was observed. This might be due to the low reproducibility of the experiments due to the non-uniform distribution of the MWCNTs. To this end, it seems that in terms of flexural strength results, the reinforcing concentration within the region of 0.1 up to 0.3 wt% essentially increased flexural strength, with 0.2 wt% being the optimal concentration for the maximum flexural strength. Compression tests results can also be seen in Figure 10b as green filled squares at the right *Y*-axis; little (0.1 to 0.3 wt%) concentration reinforcement resulted in increased compressive strength. Higher concentrations decreased the compressive strength and increased scatter, possibly due to the non-uniform, inhomogeneous MWCNT dispersion in the white mortar cementitious matrix.

### 3.4. Effect of Carboxyl Functionalization

Figure 12a,b show the typical electrical resistivity curves along with average values and standard deviation of the carboxyl functionalized MWCNT reinforcement, respectively. The results of the reference cement mortar were also added to the figure for complementarity purposes. It seems that carboxylation substantially increased the average electrical resistivity values of the specimens from 0.4 MOhm·cm to approximately 1.0 MOhm·cm, which corresponds to more than a 150% increase. Additionally, the standard deviation of the results seemed to be highly increased (results ranged from 0.90 up to 1.05 MOhm·cm), which is evidence of the poor MWCNT dispersion.

Regarding the mechanical test results, Figure 13a shows typical flexural stress–CMOD results for the plain cement mortar, the 0.2 wt% MWCNT reinforcement, and the respective carboxyl functionalized reinforcement. The 0.2 wt% MWCNT addition enhanced the mechanical performance of the specimens while the functionalization seemed to decrease the maximum flexural stress (fracture stress). This small decrease of approximate 5% in flexural strength (Figure 13b) was negligible when compared with the enormous increase in electrical resistivity values. The increased resistivity value is evidence of low MWCNT dispersion or a decrease in the electron carrying capability of the MWCNTs. In the work of Reales et al. [46], it was reported that carboxyl functionalization is often responsible for the presence of surface defects in the MWCNTs. Specifically, the treated with 65% nitric acid (HNO_3_) MWCNTs provided higher electrical resistivity because the oxidation increased the defect population on the MWCNTs due to length shortening [47]. At the same time, the existence of these high defect populations and the loss/decrease in the population of the –OH of the pristine material is the reason that the dispersion became weaker than the treated-MWCNTs in the water solution, and finally, the recorded flexural stress decreased slightly. To this end, the authors acknowledge that this functionalization treatment might cause surface defects or possible fracture of the MWCNTs, negatively affecting the electrical and mechanical properties of the white mortar.

## 4. Conclusions

In the present study, different surfactant types and a superplasticizer were examined for the efficient dispersion of multi-wall carbon nanotubes in white cement restoration mortar. The results of the study can be summarized as follows:The surfactants sodium dodecylbenzene sulfonate (SDBS) and Triton X-100 proved incompatible with cement-based composites. The surfactants caused a slight decrease in the electrical conductivity (by an average of −10%) of the matrix; however, the mechanical properties were essentially decreased by an average of 60%.The superplasticizer Ceresit CC 198 was found to be appropriate for the satisfactory dispersion of MWCNTs in the white mortar cementitious matrix.The optimal superplasticizer concentration was determined to be around 0.8 wt% of cement as it led to the electrical resistivity decrease and increase in both the flexural and compressive strength of the white cement mortar.The optimum concentration of MWCNTs was determined to be around 0.2 wt% of cement and therefore the ratio of 1 to 4 in MWCNTs to superplasticizer is appropriate.The nanocomposite reinforced with MWCNTs without carboxyl-functionalization showed better results than the MWCNTs with chemical modification, possibly due to surface defects on the MWCNTs during functionalization.

## Figures and Tables

**Figure 1 nanomaterials-12-01031-f001:**
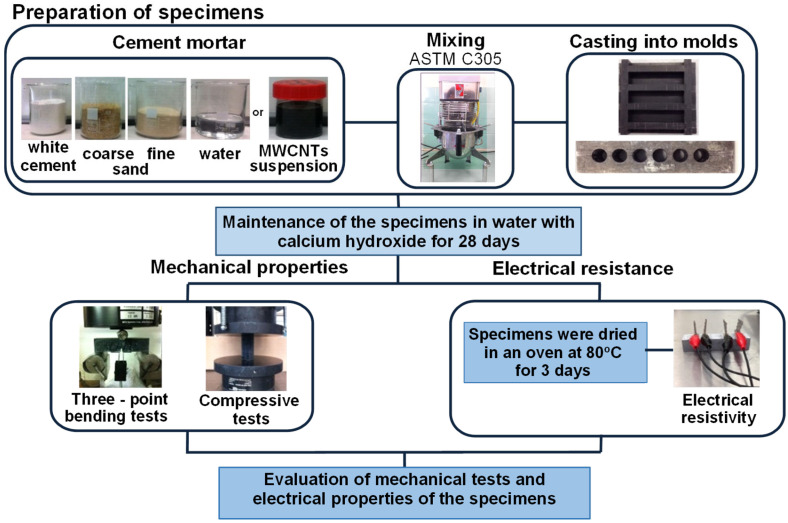
Flow diagram of the experimental procedure followed to prepare the nanocomposites and perform the experimental testing.

**Figure 2 nanomaterials-12-01031-f002:**
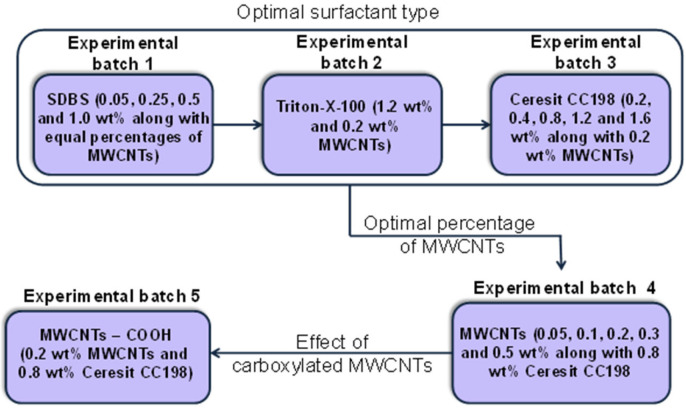
Experimental batches investigated in the present study (the percentages shown are by weight (wt%) of cement).

**Figure 3 nanomaterials-12-01031-f003:**
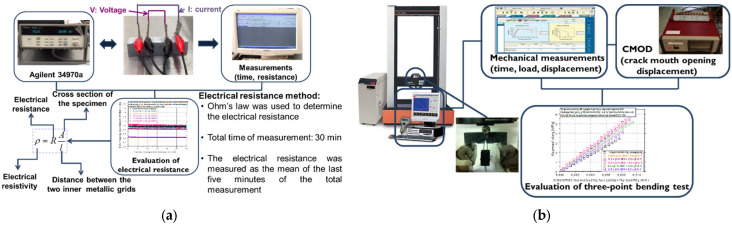
Experimental details of the (**a**) electrical resistance measurements and (**b**) flexural testing.

**Figure 4 nanomaterials-12-01031-f004:**
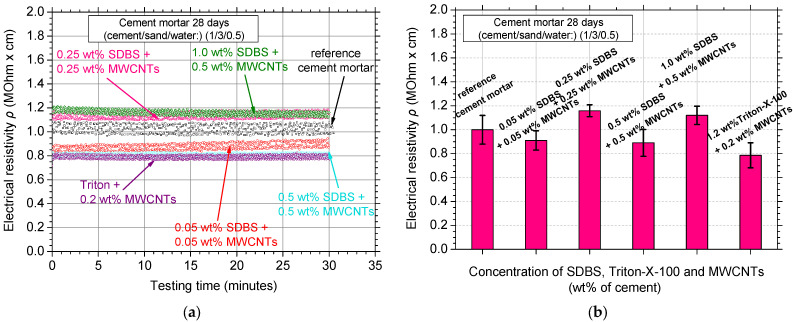
(**a**) Electrical resistivity curves over testing time and (**b**) average electrical resistivity values of the nanocomposites dispersed with the SDBS and Triton X-100 surfactants.

**Figure 5 nanomaterials-12-01031-f005:**
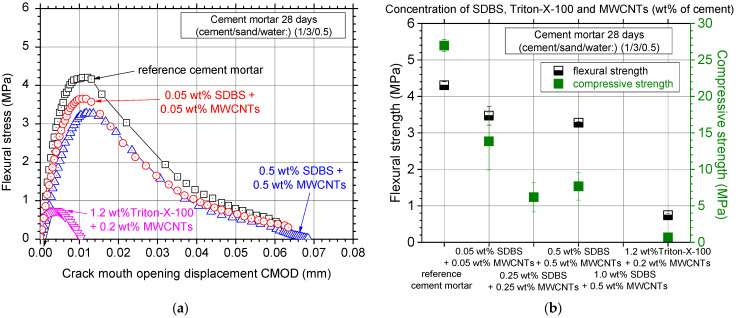
(**a**) Typical flexural stress–crack mouth opening displacement curves and (**b**) average flexural and compressive strength values and standard deviation of the nanocomposites dispersed with the SDBS and Triton X-100 surfactants.

**Figure 6 nanomaterials-12-01031-f006:**
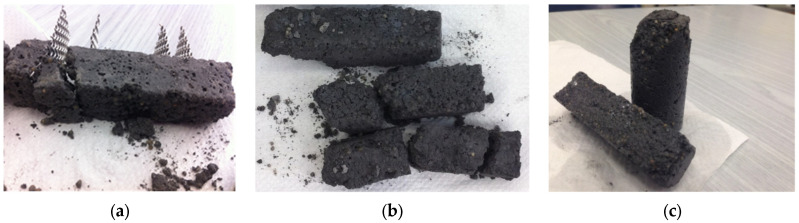
White cement mortar specimens with the Triton X-100 superplasticizer that were partially fractured during demolding: (**a**) electrical resistivity, (**b**) flexure, and (**c**) compression specimens.

**Figure 7 nanomaterials-12-01031-f007:**
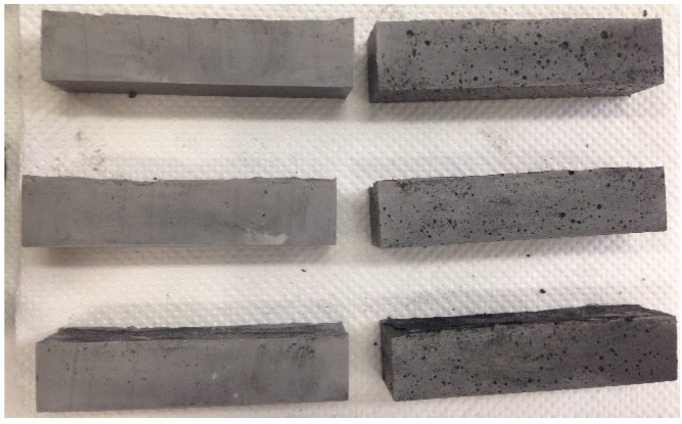
MWCNT/white cement mortar specimens dispersed with the superplasticizer on the left and the SDBS surfactant on the right.

**Figure 8 nanomaterials-12-01031-f008:**
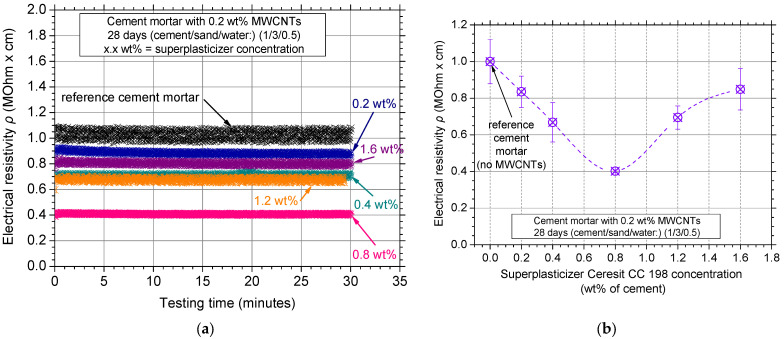
(**a**) Electrical resistivity curves over testing time and (**b**) average electrical resistivity values of the nanocomposites dispersed with the Ceresit CC 198 surfactant at different concentrations.

**Figure 9 nanomaterials-12-01031-f009:**
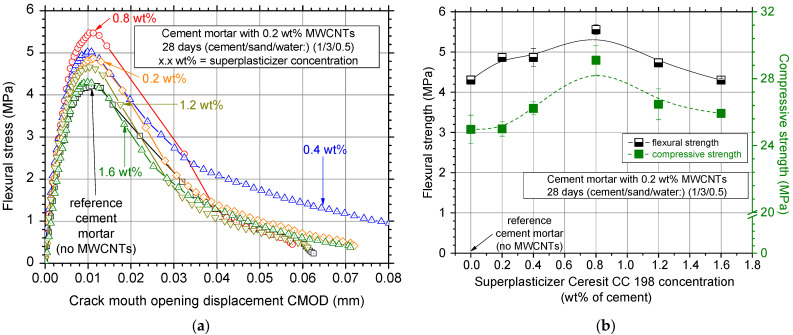
(**a**) Typical flexural stress–crack mouth opening displacement curves and (**b**) average flexural and compressive strength values and standard deviation of the nanocomposites dispersed with the Ceresit CC 198 surfactant at different concentrations.

**Figure 10 nanomaterials-12-01031-f010:**
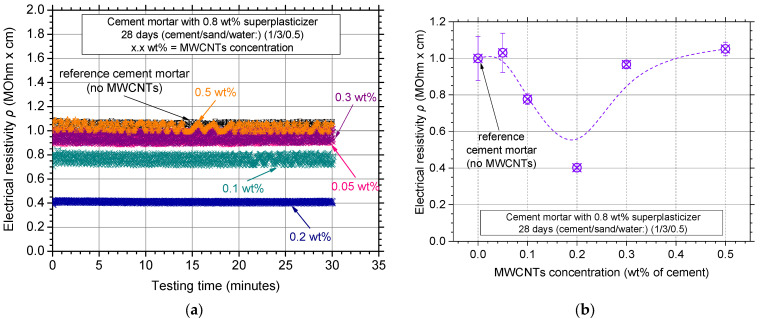
(**a**) Electrical resistivity curves over testing time and (**b**) average electrical resistivity values of nanocomposites with different MWCNT concentrations.

**Figure 11 nanomaterials-12-01031-f011:**
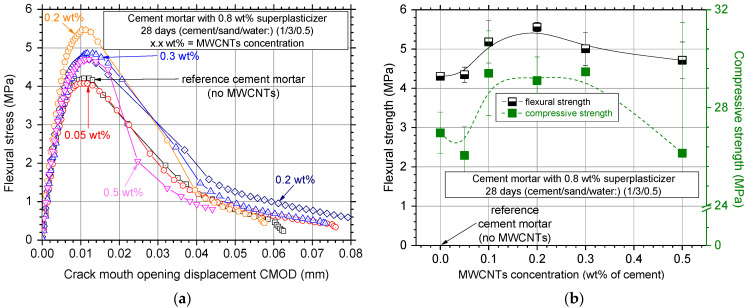
(**a**) Typical flexural stress–crack mouth opening displacement curves and (**b**) average flexural and compressive strength values and standard deviation of nanocomposites with different MWCNT concentrations.

**Figure 12 nanomaterials-12-01031-f012:**
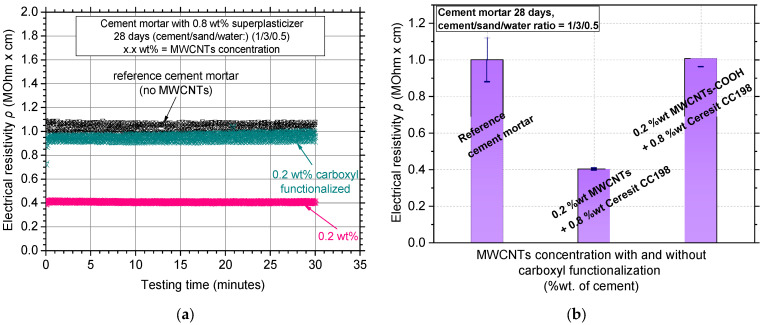
(**a**) Electrical resistivity curves over testing time and (**b**) average electrical resistivity values of MWCNT/white mortar nanocomposites where MWCNTs were treated with and without carboxyl functionalization.

**Figure 13 nanomaterials-12-01031-f013:**
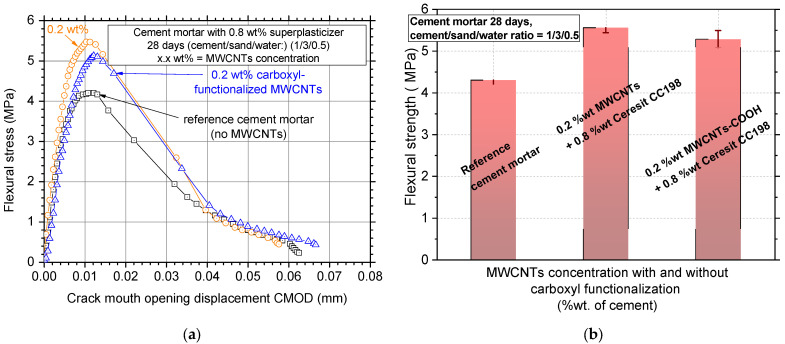
(**a**) Typical flexural stress–crack mouth opening displacement curves and (**b**) average flexural strength values and standard deviation of MWCNT/white mortar nanocomposites where MWCNTs were treated with and without carboxyl functionalization.

**Table 1 nanomaterials-12-01031-t001:** Mineralogical phases of white cement.

C3S	C2S	C3A	C4AF
77 (wt%)	16 (wt%)	5 (wt%)	1 (wt%)

**Table 2 nanomaterials-12-01031-t002:** Physical characteristics of white cement.

Density (kg/m^3^)	3130
Phenomenological density (kg/m^3^)	1100
Curing time of cement according ΕΝ 196-3	120 min

**Table 3 nanomaterials-12-01031-t003:** Mixing quantities (g) of the materials used for the MWCN/aqueous dispersions.

Variance in the Surfactant/Nano-Reinforcement	MWCNTs (g)	SDBS (g)	TRITON (g)	CERESIT (g)	WATER (g)
**SDBS-1**	0.5	0.5	-	-	500
**SDBS-2**	2.5	2.5	-	-	500
**SDBS-3**	5	5	-	-	500
**SDBS-4**	5	10	-	-	500
**Triton-X**	2	-	12	-	500
**Ceresit-1**	2	-	-	4	500
**Ceresit-2**	2	-	-	6	500
**Ceresit-3**	2	-	-	8	500
**Ceresit-4**	2	-	-	12	500
**Ceresit-5**	2	-	-	16	500
**MWCNTs-1**	0.5	-	-	8	500
**MWCNTs-2**	1	-	-	8	500
**MWCNTs-3**	2	-	-	8	500
**MWCNTs-4**	3	-	-	8	500
**MWCNTs-5**	5	-	-	8	500
**MWCNTs-6**	2 *	-	-	8	500

*—COOH functionalized.

**Table 4 nanomaterials-12-01031-t004:** Mixing ratios of the materials used for mortars.

	Cement (g)	Quartz Sand		*w*/*c*
		Coarse (g)	Fine (g)	
Mortar	260	520	260	0.5

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
