# Peer review of "Dispersion of Multi-Walled Carbon Nanotubes into White Cement Mortars: The Effect of Concentration and Surfactants"

_nanomaterials, 2022, doi:10.3390/nano12061031_

Round 1

Reviewer 1 Report

The manuscript “Dispersion of multi-walled carbon nanotubes into white cement mortars: the effect of concentration and surfactants” addresses the influence of surfactants and MWCNTs addition on the electric and mechanical properties of white cement mortars. Effects of type of surfactant and MWCNT concentration are studied. The presented work is simple but I consider that the conclusions are important and well supported by the data. Surprisingly, the surfactants have a negative effect in the conductivity (slightly) and mechanical properties so cannot be employed for this purpose. I miss further discussion on this point. Another important conclusion is that -COOH functionalized MWCNTs also behave poorly in comparison with as raw ones and here is my main concern about this manuscript.

There are no data about the characteristics of the MWCNTs (apart from the typical commercial information). I couldn’t find information on the functionalization treatment to oxidize the MWCNTs. At least, it is necessary to present a surface analysis of both types of MWCNTs (ideally by XPS), so the amount of oxygen and type of oxygen groups in both samples can be derived (from the C 1s high resolution peak, for example). Otherwise it is impossible to draw out any conclusions between the different behavior between non-oxidized/oxidized samples (it is well known that oxidized MWCNTs don’t present -COOH groups only).

From my point of view, I consider that this work is worth of publication in Nanomaterials if a bit more effort is placed on the characterization of the nanotubes. 

Reviewer 2 Report

The paper is focused on the influence of surfactants and plasticizers on the dispersibility of multi-walled carbon nanotubes into white cement mortars. The presentation and discussion of the results are poor. I recommend the publication after the following major revisions:

  • Table 3. The unit for gram should be indicated as “g” instead of “G”.
  • Morphological investigations of the reinforced cement-based materials should be added.
  • English language should substaintally improved.
  • A better comparison with literature results should be carried out.
  • In several parts, the authors do not provide any scientific explanation on the obtained results (for instance, see lines 281-290). The discussion on the experimental data should be significantly improved.

Round 2

Reviewer 1 Report

The reference cited to clarify my major concern is not adequate as it is related to phenol moieties attachment by p-aminophenol reactants. Treatments are not comparable. Moreover, there is no XPS characterization in that reference, so it is impossible to know where the new added table of XPS data comes from.

Furthermore, the MWCNTs in the cited paper seem to be laboratory synthesized while those in the present manuscript are claimed to be commercial. Therefore, the raw material is nor comparable either.

The XPS data are extremely strange as a concentrated nitric acid treatment increases the sp2 /sp3 ratio. This is unexpected and contrary with the increase in defect population that authors use to justify the higher electrical resistivity of the treated MWCNTs.

None of this help to clarify my comments.  

Reviewer 2 Report

The paper can be published in the present form.

Author Response

We thank the reviewer for his/her kind comments